# Potential Mechanisms of Plant-Derived Natural Products in the Treatment of Cervical Cancer

**DOI:** 10.3390/biom11101539

**Published:** 2021-10-18

**Authors:** Meizhu He, Lijie Xia, Jinyao Li

**Affiliations:** Xinjiang Key Laboratory of Biological Resources and Genetic Engineering, College of Life Science and Technology, Xinjiang University, Urumqi 830017, China; hmz1224@stu.xju.edu.cn

**Keywords:** plant-derived natural products, cervical cancer, molecular mechanisms, treatment

## Abstract

Cervical cancer is the second most common gynecological malignancy globally; it seriously endangers women’s health because of its high morbidity and mortality. Conventional treatments are prone to drug resistance, recurrence and metastasis. Therefore, there is an urgent need to develop new drugs with high efficacy and low side effects to prevent and treat cervical cancer. In recent years, plant-derived natural products have been evaluated as potential anticancer drugs that preferentially kill tumor cells without severe adverse effects. A growing number of studies have shown that natural products can achieve practical anti-cervical-cancer effects through multiple mechanisms, including inhibition of tumor-cell proliferation, induction of apoptosis, suppression of angiogenesis and telomerase activity, enhancement of immunity and reversal of multidrug resistance. This paper reviews the therapeutic effects and mechanisms of plant-derived natural products on cervical cancer and provides references for developing anti-cervical-cancer drugs with high efficacy and low side effects.

## 1. Introduction

Cervical cancer is one of the second most common gynecologic malignancies in the world, with more than 570,000 women diagnosed with cervical cancer and 311,000 deaths annually worldwide [1]. However, persistent high-risk human papillomavirus (HPV) infection is the primary causative agent of cervical carcinogenesis, especially HPV16 and HPV18, which cause 70% to 72% of invasive cervical cancers. HPV contains two oncoproteins (E6 and E7) that play an essential role in the cervical cancer development and progression [2,3]. Currently, the main treatments for cervical cancer include surgical resection, radiotherapy, local targeted therapy and immunotherapy. Although traditional therapies are effective for early stage cervical cancer, they have limited efficacy for locally staged and metastatic cervical cancer due to severe side effect, drug resistance, multiple recurrences and metastases [4,5]. Therefore, there is an urgent need to develop drugs with a better safety profile and higher efficacy for cervical cancer treatment.

In recent years, plant-derived natural products have been considered as the most promising candidates for oncology therapies. They can preferentially kill tumor cells with low toxic effect on normal cells. This is attributed to their chemical diversity, structural complexity, inherent biological activity and low side effects [6,7]. A variety of plant-derived natural products with antitumor activities have been identified, such as flavonoids, terpenoids, alkaloids and phenols, which can inhibit tumor-cell proliferation, induce apoptosis, reduce telomerase activity, suppress angiogenesis, improve immune function, reverse multidrug resistance, etc. [8,9,10]. In this paper, we review the recent studies on the role and mechanisms of plant-derived natural products in the treatment of cervical cancer.

## 2. Methods

We collected the relevant experimental works in the literature published in the last five years that elucidate the anticancer effects of natural products on cervical cancer, using PUBMED (including Medline) and the Google Scholar database. Upon searching for the appropriate studies, we used “natural product, cervical cancer” as keywords. After completing the initial search, we removed duplicate works from the literature. Chemical structures of compounds derived from natural products were cited from the NCBI PubChem website, available online: https://pubmed.ncbi.nlm.nih.gov/ (accessed on 10 June 2021).

## 3. Flavonoids

Flavonoids are phenolic phytochemicals that are commonly found in fruits, vegetables and plant-based beverages (e.g., green tea and wine) [11]. More than 8000 species have been identified and isolated from plants. Their structures consist of C_6_-C_3_-C_6_ skeletons labeled with A, B and C rings, which can be classified into flavones, flavanones, flavonols, flavanols, isoflavones and anthocyanidins based on their structural diversity [12,13]. Many studies have reported that flavonoids have a significant role in tumor prevention and treatment attributed to their wide range of biological activities, such as anti-inflammatory, antioxidant, anti-hyperlipidemic, anti-fatigue, anti-aging, etc. [14,15,16]. In addition, they can induce apoptosis of tumor cells by inhibiting various pro-cancer pathways and genes in tumor cells.

### 3.1. Flavones

*Scutellaria baicalensis* is one of the versatile herbs traditionally and has been used in China to treat inflammatory diseases, hypertension, cardiovascular diseases, bacterial and viral infections. A large amount of evidence suggests that *S. baicalensis* also has potent anticancer activity, and its main bioactive components are baicalein, Wogonin, and baicalin [17,18]. Baicalein is a flavonoid derived from the roots of *S. baicalensis* and has a variety of pharmacological activities, which can inhibit cell proliferation and migration; induce apoptosis and cell-cycle arrest [19]. Cyclin D1 is a potential therapeutic target for cervical cancer. Baicalein inhibits cyclin D1 overexpression and arrests the cell cycle at G0/G1 phase through Wnt/β-catenin and protein kinase B/glycogen synthase kinase-3β (AKT/GSK-3β) signaling pathways to suppress proliferation and induce apoptosis of human cervical cancer HeLa and SiHa cells [20,21]. According to reports, baicalein inhibited tumor necrosis factor alpha (TNF-α)-induced the activation of nuclear factor kappa-light-chain-enhancer of activated B cells (NF-κB) and the expression of apoptosis protein 1 (cIAP-1), cIAP-2, FLIP, B-cell lymphoma 2 (Bcl-2), matrix metalloproteinase 2 (MMP2), MMP9, first apoptosis signal receptor (Fas), FasL, caspase 8 and vascular endothelial factor (VEGF) in a dose-dependent manner to suppress HeLa cell invasion and migration. On the other hand, baicalein blocked TNF-α-induced nuclear translocation of p65 through inhibiting the phosphorylation and degradation of the inhibitory subunit of NF-κB (IκBα), activated caspase 8 and promoted the cleavage of poly (ADP-ribose) polymerase (PARP) expression to induce apoptosis in HeLa cells. In addition, baicalein had strong anti-inflammatory activity by inhibiting the phosphorylation of extracellular signal-regulated kinases 1/2 (ERK1/2) and p38, which reduced the expression of the inflammatory cytokines, including interleukin 8 (IL-8) and monocyte chemoattractant protein 1 (MCP1) [22,23,24]. Moreover, baicalein inactivated the AKT/mammalian target of rapamycin (mTOR) pathway by targeting the circHIAT1/miR-19a-3p axis to inhibit the proliferation of cervical cancer cells [25]. Another study illustrated that baicalin significantly suppressed cervical cancer xenograft tumor growth and metastasis in vivo through intraperitoneally injecting 10 mg/kg/d baicalin for four weeks. It was also reported that baicalein downregulated the expression of long non-coding RNA (lncRNA) in a dose- and time-dependent manner and named baicalein down-regulated long non-coding RNA (BDLNR). As it physically bound itself to Y-box binding protein 1 (YBX1), recruited YBX1 to the PIK3CA promoter, and it mediated the anti-cancer effects of baicalein in cervical cancer via activating PI3K/Akt pathway [26]. It is suggested that BDLNR may be a potential therapeutic target to enhance the anticancer effect of baicalein.

Wogonin is a natural monoflavonoid with the potential for selective tumor therapy in vitro and in vivo [27]. Wogonin could decrease the expression of HPV oncoproteins E6 and E7 and induce apoptosis in SiHa and CaSki cells through the mitochondria-mediated pathway, thus reducing mitochondrial membrane potential (MMP); elevating the Bcl-2-associated X protein (Bax)/Bcl-2 expression ratio; leading to cytochrome c (Cyt c) release; and triggering the cleavage of caspase 3, caspase 9 and PARP [28]. Wogonin also inhibited HeLa cell proliferation by inducing G1-phase cell-cycle arrest and apoptosis via decreasing the expression of cyclin D1, cyclin-dependent kinase 4 (CDK 4), pRb and nuclear transcription factor E2F-1, as well as increasing the expression of cyclin-dependent kinase inhibitor 1A/CDK-interacting protein 1(p21cip1) at the mRNA and protein levels through a p53-dependent mechanism [29]. Moreover, wogonin can enhance the effect of cisplatin on the induction of cancer cell apoptosis through reactive oxygen species (ROS)-dependent mechanism [30]. The results indicate that wogonin is a potential anticancer agent.

Apigenin is an edible natural flavonoid found in various dietary plant foods, such as vegetables and fruits [31]. It has a strong inhibitory effect on tumor-cell viability in vivo and in vitro. Its derivative apigenin 7-glucoside has anti-inflammatory, antioxidant and anticancer activities [32]. Apigenin 7-glucoside significantly suppressed the proliferation of HeLa cells, and the IC_50_ value at 48 h was 47.26 μmol/L. It induced apoptosis in HeLa cells through the death receptor pathway and the mitochondrial pathway, which effectively increased the expression of ROS, Fas, FasL, TNF-α, TNF-r1, Fas-associated death domain (FADD), TNF receptor-associated death domain (TRADD), caspase 3 and caspase 9 and decreased the expression of pro-caspase 8, caspase 10, Bcl-2 and Bcl-2 extra-large protein (Bcl-xl). Additionally, apigenin 7-glucoside treatment increased p16 INK4A expression and reduced Cyclin (A, D, E) and CDK2/6 expression. Meanwhile, it inhibited HeLa cell migration by targeting the phosphatase and tensin homolog (PTEN)/PI3K/AKT pathway [33]. Some studies reported that apigenin showed selective sensitivity for human cervical cancer HeLa, SiHa, CaSki and C33A cells with IC_50_ values of 10, 68, 76 and 40 μmol/L, respectively. Apigenin also induced mitochondrial redox damage, decreased MMP and lipid peroxidation and inhibited the migration and invasion of cervical cancer cells [34]. Zhang et al. found that Apigenin suppressed cervical tumor growth in vivo by attenuating histamine-induced abnormal estrogen receptor signaling by increasing the estrogen receptor β/estrogen receptor α (ERβ/ERα) ratio when tumor model mice were administered intraperitoneally with 100 mg/kg apigenin +1 mg/kg histamine every 3 d. Moreover, apigenin induced autophagy and apoptosis in HeLa cells through the PI3K/AKT/mTOR signaling pathway [35]. Consequently, apigenin and its derivatives may prevent the development and progression of cervical cancer and represent promising drugs for cervical cancer therapy.

Luteolin is a common plant flavonoid found in various plants, including fruits, vegetables and medicinal herbs. Luteolin-rich plants have been used to treat multiple diseases, such as hypertension, cancer and inflammation [36,37]. Luteolin could dose-dependently reduce the proliferation of HeLa cells with IC_50_ value of 21.8 μmol/L. Luteolin induced apoptosis and G2/M phase cell-cycle arrest in HeLa cells by downregulating UHRF1 and DNA methyltransferase 1 (DNMT1) with a reduction of overall DNA methylation and upregulating p16 INK4A expression [38]. On the other hand, luteolin promoted apoptosis by inhibiting TNF-α-induced NF-κB activation, downregulating A20 and c-IAP1 gene expression and enhancing c-Jun N-terminal kinase (JNK) activity [39]. It was also reported that luteoloside inhibited HeLa cell proliferation by endogenous and exogenous pathways, which increased the ratio of Bax/Bcl-2 to reduce MMP and release Cyt c, upregulated Fas expression and activated caspase 8 and caspase 3. At the same time, luteoloside inhibited mTOR and activated p38 mitogen-activated protein kinase (MAPK) signaling pathways to exert anti-cervical-cancer effects [40]. In addition, luteolin combined with TNF-related apoptosis-inducing ligand (TRAIL) synergistically induced apoptosis in HeLa cells through upregulation of death receptor 5 (DR5) and Bid cleavage, and activation of caspase-8 [41]. These data suggest that luteolin may serve as a new therapeutic strategy for cervical cancer.

### 3.2. Flavanones

Citrus fruits are rich in flavonoids and are known for their health-promoting and chemopreventive properties. Naringin, a flavonoid glycoside, can be isolated from citrus fruits, such as orange, tangerine, lemon and lime, and has several pharmacological activities [42]. Naringin can significantly reduce cancer cell viability and proliferation without toxicity to normal tissue cells [43]. The IC_50_ values of naringin for HeLa, SiHa and C33A cells after 24-h treatment were 793, 764 and 745 μmol/L, respectively, and showed a dose-dependent relationship. Naringin increased the level of endoplasmic reticulum (ER) stress sensors, phosphorylated eIF2α and activated the apoptosis-related protein CHOP and other associated pro-apoptotic proteins (PARP1). Importantly, naringin also blocked the β-catenin signaling pathway by decreasing β-catenin (Ser576) and GSK-3β (Ser9) protein expression and phosphorylation and induced cell-cycle arrest at G0/G1 phase by increasing the expression of cell-cycle checkpoint proteins p21/cip and p27/kip to trigger apoptosis in cervical cancer cells [44]. Another study reported that naringin promoted the expressions of caspases (3, 8 and 9), p53, Bax, Fas death receptor and its adaptor protein FADD to induce apoptosis in HeLa cells through the death receptor pathway and the mitochondrial pathway. The activation of the exogenous pathway was associated with increased caspase 8, which activated the death receptor by cleaving Bid into tBid and the crosstalk between the death receptor and mitochondrial pathways [45]. Moreover, naringin also induced growth inhibition and apoptosis in HeLa cells by decreasing the expression of NF-κBp65, COX-2 and caspase 1 [46]. The results suggest that naringin is a potentially effective drug for treating human cervical cancer.

Hesperidin is a citrus flavonoid and has various functions, including antioxidation, antitumor and anti-angiogenesis functions, which can protect mitochondrial membranes from free radical attack and inhibit tumor-cell migration and invasion [47,48]. Hesperetin inhibited the cell viability of SiHa cells in a dose- and time-dependent manner with IC_50_ values of 650 μmol/L. Hesperetin induced cell-cycle arrest at G2/M phase and apoptosis in SiHa cells by death receptor and mitochondrial pathways, characterized by depolarizing MMP and increasing the expression of caspase 3, caspase 8, caspase 9, p53, Bax, Fas and FADD [49]. In addition, hesperidin inhibited HeLa cell proliferation and induced apoptosis through ER stress and mitochondria-mediated pathways via decreasing the expression of cyclin D1, cyclin E1 and CDK2, promoting Cyt c release, and activating the expression of apoptosis-inducing factor (AIF), GADD153/CHOP and glucose-regulated protein of 78 kDa (GRP78) [50]. Hesperidin has potential in preventing and treating cervical cancer and may open new avenues for cancer treatment.

*Silibinin* is a bioactive polyphenolic flavonoid isolated from the fruits and seeds of *silybum marianum*. It has been used to treat various diseases, especially the liver, gallbladder and kidney [51,52]. *Silibinin* inhibited the growth of Hela and SiHa cells in a dose- and time-dependent manner, with IC_50_ values of 332 and 275 μmol/L for HeLa cells, and 250 and 195 μmol/L for SiHa cells at 48 h and 72 h, respectively. It reduced ATP content, mtDNA copy number and MMP, activated the dynamin-related protein 1(Drp1)-mediated mitochondrial fission pathway, induced G2/M cell-cycle arrest through reducing the expression of CDK1, cyclin B1 and Cdc25C [53]. S*ilibinin* also suppressed angiogenesis and promoted apoptosis in HeLa cells through inhibiting the mTOR/p70S6K/4E-BP1 signaling pathway and decreasing the accumulation and transcriptional activity of hypoxia-inducible factor-1α (HIF-1α) and the release of VEGF. Moreover, the anti-angiogenic ability of s*ilibinin* was enhanced by blocking AKT activation through PI3K/AKT inhibitor LY294002 [54]. At the same time, s*ilibinin* induced apoptosis in HeLa cells through the mitochondrial pathway and the death-inducing pathway, resulting in decreased expression of CDK1 and CDK2 proteins and increased ratio of Bax/Bcl-2, followed by Cyt c release and activation of caspase 9, as well as increased expression of Fas and FasL and activation of caspase 8 [55]. The evidence has shown that hydroxyl radical (-OH) is the main form of s*ilibinin*-induced ROS [56]. S*ilibinin* inhibits HeLa cell growth by activating apoptotic vesicles and caspase 3, and promoting the phosphorylation of p53 and JNK in a dose-dependent manner. p53 subsequently interferes with mitochondrial function through the p53-upregulated modulator of apoptosis (PUMA) pathway, which upregulates the Bax/Bcl-2 ratio, reduces MMP and increases ROS production. Then ROS induces autophagy and apoptosis in HeLa cells. Furthermore, p53-mediated glutathione (GSH) depletion significantly enhanced the cytotoxicity of NO in HeLa cells [57,58]. It may be a classical candidate for the design of anticancer drugs.

### 3.3. Flavonols

Kaempferol is a natural flavonol and widely distributes in many plant families. The growing evidence has shown that kaempferol is a potential cancer therapeutic agent with potent antitumor, anti-inflammatory and antioxidant properties [59]. It inhibited SiHa cell growth and proliferation in a dose- and time-dependent manner, and promoted SiHa cell apoptosis by disrupting MMP and elevating intracellular free Ca^2+^ concentration, which caused shrinkage of spindle-shaped SiHa cells and damage of their microtubule networks [60]. Another study showed that Kaempferol inhibited HeLa cell growth in a time- and concentration-dependent manner and the IC_50_ value was 10.48 μmol/L at 72 h, while it had weak toxicity for normal cells with IC_50_ value of 707 μmol/L at 72 h. It induced apoptosis and senescence in HeLa cells by inhibiting PI3K/AKT pathway and human telomerase reverse transcriptase (hTERT) expression and promoting the p53 pathway [61]. Telomerase is considered a new and potentially selective target for tumor therapy. Telomerase inhibition by kaempferol may provide a safe and effective approach for the treatment of cervical cancer. Additionally, overexpression of P-glycoprotein (P-gp) causes efflux of chemotherapeutic drugs from cells and is considered to be one of the crucial mechanisms of multidrug resistance (MDR) in cancer [62]. It was reported that kaempferol significantly decreased the activity and function of P-gp in MDR human cervical cancer KB-V1 cells in a dose-dependent manner, reduced drug efflux, improved sensitivity to chemotherapeutic drugs vinblastine and paclitaxel as well as cytotoxicity, which in turn induced apoptosis and reversed MDR in KB-V1 cells [63]. Kaempferol might be a potential candidate for the prevention and treatment of cancer due to its safety and low-cost advantages.

Quercetin is a dietary polyphenolic compound with wide distribution in vegetables and fruits and has various activities, including anti-allergic, anti-inflammatory and antitumor activities [64,65]. Quercetin can suppress tumor-cell proliferation and induce apoptosis, cell-cycle arrest and DNA damage through intrinsic apoptosis pathway, which involved in PI3K, MAPK and Wnt [66]. Priyadarsini et al. showed that quercetin could target both opposing signaling pathways, p53 and NF-κB to inhibit the proliferation of HeLa cells [67]. Quercetin induced apoptosis of HeLa and SiHa cells by blocking the interaction of E6/E6AP complex, reactivating p53 and upregulating the expression of transcriptional target p21 [68]. Further, quercetin promoted apoptosis and reduced migration of HeLa cells within 18 h by downregulating the expression of AKT and Bcl-2, and blocked the cell cycle at the G2/M phase. The accumulation of ROS increased Cyt c release and MMP depolarization and activated caspase 3, which in turn exhibited significant anti-proliferative and pro-apoptotic effects on HeLa cells [69]. Interestingly, the combination of quercetin with other chemical agents effectively enhanced the antitumor effect. Quercetin inhibited the viability of HeLa and SiHa cells in a dose- and time-dependent manner, with an IC_50_ of 30 μmol/L for HeLa cells at 24 h and 50 μmol/L for SiHa cells at 48 h. Treatment with cisplatin or quercetin alone did not reduce the expression of MMP2 protein in cervical cancer cells but their combination significantly decreased MMP2 expression, which inhibited the migration and invasion of cervical cancer cells. Meanwhile, quercetin enhanced the chemosensitivity of cervical cancer cells by downregulating the expression of P-gp and methyltransferase-like 3 (METTL3), which mediated HeLa cell proliferation and apoptosis [70]. Quercetin also increased the sensitivity of HeLa cells to cisplatin by inhibiting the expression of the multidrug resistance-associated protein (MRP) and heat shock protein Hsp72, which induced apoptosis and reversed cellular resistance [71]. These studies provided the experimental basis for the treatment of cisplatin-resistant patients.

Fisetin is a natural flavonols found in vegetables, fruits and nuts and has antitumor, anti-invasive, anti-angiogenic, antidiabetic, cardioprotective and neuroprotective activities [72,73]. Fisetin suppressed urokinase-type plasminogen activator (u-PA) expression by blocking the phosphorylation of p38 MAPK and the nuclear translocation of NF-κB, which inhibited the migration and invasion of SiHa cells. Moreover, the addition of the p38 MAPK inhibitor SB203580 further enhanced the inhibitory effect of fisetin on u-PA activity and expression [74]. It was also reported that fisetin not only promoted apoptosis protease activating factor-1 (Apaf-1) expression and Cyt c release to activate caspase 3 and caspase 9 but also inhibited ERK1/2 phosphorylation, COX-2 expression and prostaglandin E2 (PGE2) production by blocking the NF-κB/p300 signaling pathway, which in turn promoted HeLa cell apoptosis [75]. Furthermore, fisetin significantly inhibited the growth rate of tumors with inhibition rates of 82.65% and 92.62% in a mouse model of HeLa cell line injection without significant side effects [76]. It was also experimentally proven that fisetin combined with sorafenib could activate the DR5-mediated death receptor pathway and mitochondria-dependent pathway, which upregulated Bax/Bcl-2 ratio and promoted MMP depolarization, caspase 3/caspase 8 activation and PARP cleavage to induce apoptosis in HeLa cells [77]. Fisetin is expected to be an anticancer drug for the clinical treatment of cervical cancer.

### 3.4. Flavanols

Green tea (*Camellia sinensis*) is one of the most commonly used herbs globally and is widely known for its effectiveness in preventing chronic diseases and tumors. This is mainly attributed to the biologically active catechin compounds in green tea, such as (−)-epigallocatechin gallate (EGCG), (−)epigallocatechin 3-gallate (ECG), (−)epigallocatechin (EGC) and (+) catechin. Among them, EGCG is the main component of catechins and has substantial antioxidative, antitumor and anti-angiogenic effects [78,79]. After treatment with 150 and 300 μg/mL of catechin for 72 h, apoptosis rates of SiHa cells reached 31.62% and 34.8%, with an IC_50_ value of 196.07 μg/mL. In addition, catechin inhibited the proliferation and induced apoptosis of SiHa cells partly by regulating TP53 and caspase 3, caspase 8 and caspase 9 [80]. EGCG concentration- and time-dependently inhibited HeLa cell proliferation with an IC_50_ value of 20 µg/mL. EGCG inhibited the activation of AKT and NF-κB by blocking the phosphorylation and degradation of the inhibitory κBα and κBβ subunits to result in the downregulation of COX-2 expression. Furthermore, EGCG treatment led to mitochondrial dysfunction through increasing ROS production, p53 and Bax/Bcl-2 ratios to promote Cyt c release and activation of caspase cascade, which induced apoptosis in HeLa cells [81]. Recently, one study indicated that EGCG inhibited transforming growth factor-β (TGF-β)-induced epithelial-mesenchymal transition (EMT) in Hela and SiHa cells via the ROS/Smad signaling pathway to inhibit cell migration and invasion [82]. Besides, EGCG can be used as an anti-angiogenic agent in the treatment of cervical cancer. EGCG significantly suppressed hypoxia and serum-induced accumulation of HIF-1a protein, as well as the expression of VEGF by blocking PI3K/Akt/mTOR and ERK1/2 signaling pathways, thereby inhibiting HeLa cell angiogenesis [83]. Meanwhile, EGCG enhances the sensitivity of cisplatin to cervical cancer cells by inhibiting the mTOR signaling pathway and the levels of p-p70S6K1 and p-4E-BP1, which in turn inhibits cell viability and induces apoptosis in HeLa cells [84]. It has also been shown that EGCG can induce apoptosis in HeLa cells by inhibiting telomerase activity [85]. It is suggested that telomerase inhibition may be one of the critical mechanisms of EGCG treatment.

### 3.5. Isoflavones

Genistein is considered the primary isoflavone in soy foods and is one of the most widely studied phytoestrogens in the Asian diet [86]. It was demonstrated that oral administration of 20 mg/kg of genistein exerted anti-proliferative and immunomodulatory effects in cervical cancer mouse model, which promoted lymphocyte proliferation and lactate dehydrogenase (LDH) release, enhanced cytolytic activity and IFN-γ production, and thus induced protective antitumor immunity [87]. AKT signaling pathway plays a crucial role in controlling cell survival and apoptosis. Genistein enhanced the activity of cisplatin by inhibiting the expression of the AKT/mTOR pathway, p-p70S6K1 and p-4E-BP1, which led to the growth inhibition of cervical cancer cells [88]. Several studies indicated that inhibition of ERK or PI3K signaling pathways, along with activation of the p38-JNK pathway, shifted the balance toward apoptosis to result in higher levels of growth inhibition in a variety of tumor-cell lines [89,90]. Kim et al. found that genistein effectively inhibited the growth of HeLa and CaSki cells with IC_50_ values of 20 and 60 μmol/L after 48 h. It inhibited cervical cancer cell proliferation by attenuating ERK1/2 activity and activating p38 MAPK, where AKT and JNK were partially involved in genistein-induced cancer cell growth inhibition [91]. In previous study, phytoestrogens have also been found to have an anti-tumor effect [92]. Genistein blocked the estrogen receptor-mediated PI3K/AKT-NF-κB signaling pathway and exerted antitumor effects by downregulating Bcl-2, VEGF, and tumor expression necrosis factor-associated receptor factor 1 (TRAF1) and promoting apoptosis in HeLa cells [93]. Genistein also induced apoptosis in HeLa cells via a p53-dependent pathway that disrupted the interaction between p53 and APE1 to increase the intracellular stability of p53 [94]. Therefore, genistein may have potential clinical application in the treatment of cervical cancer.

Puerarin is the main active ingredient extracted from *Pueraria tuberosa*, which has beneficial effects on cardiovascular diseases, neurological dysfunction, osteoporosis, liver injury and inflammation. Puerarin could effectively inhibit the production of pro-inflammatory cytokines in a variety of disease models [95]. Puerarin also inhibited tumor-cell proliferation in a dose- and time-dependent manner through inducing apoptosis and blocking cellular cyclin D1 expression by PI3K/AKT/NF-κB signaling pathway [96]. Moreover, puerarin gavaged with 500 mg/kg for 15 d could increase the level of IL-2 and superoxide dismutase (SOD) activity in the plasma of U14 cervical cancer mice to enhance the immune function and antioxidant enzyme activity to scavenge excessive free radicals and reduce the damage of ROS in the body, thus achieving the antitumor effect [97]. Another study proved that puerarin (12.5–50 μmol/L) could inhibit HeLa cell proliferation and induce apoptosis through suppressing the β-catenin/Wnt/p53 signaling pathway [98]. Puerarin also significantly inhibited the expression of p-mTOR protein and activated caspase 3/9 in HeLa cells to reduce their proliferation and migration [99]. Thus, puerarin is expected to be a new drug for the prevention and treatment of cervical cancer.

Formononetin is a 7-hydroxy-40-methoxy herbal isoflavone isolated from *Pycnanthus angolensis*, red clovers, chickpea and other plants, and it possesses a variety of pharmacological activities, including antioxidant, chemopreventive, anti-inflammatory, anti-allergic, antibacterial and cardio-preventive effects [100,101]. According to the research, formononetin might mediate MRP (MRP1 and MRP2) inhibition through ROS while activate the mitochondrial apoptosis pathway, which resulted in MMP loss, increased Bax/Bcl-2 ratio and activation of caspase 3 and caspase 9. On the other hand, formononetin could also enhance the cytotoxicity of epirubicin in HeLa cells through the death receptor/caspase 8 apoptotic pathway [102]. Apart from this, formononetin inhibited HeLa cell proliferation in a concentration-dependent manner with an IC_50_ value of 72.112 ± 5.671 µmol/L. Multi-walled carbon nanotubes are the best carrier of formononetin, and their combination can further enhance ROS-mediated mitochondrial dysfunction [103]. Moreover, oral gavage of 10 mg/kg of formononetin for 31 d significantly inhibited tumor growth in BALB/c mouse model inoculated with HeLa cells and reduced the expression of HIF-1α and VEGF in tumor tissues without severe side effects [104]. Moreover, formononetin (oral gavage of 20 and 40 mg/kg for 35 d) activated caspase 3 and upregulated the expression of Bax by inhibiting the transduction of PI3K/AKT signal pathway, accompanied by the release of Cyt c, and then induced apoptosis of HeLa cells [105]. It might be a potential drug for cervical cancer treatment.

### 3.6. Anthocyanins

Anthocyanins are natural pigments in the plant kingdom and endow plants their blue color and antioxidant potential. Anthocyanins are mainly found in small berries, such as strawberries, blueberries, cranberries, cherries and black raspberries. More than 600 structurally diverse anthocyanins have been identified in nature [106]. Cyanidin 3-O-glucoside is the main active component of anthocyanins, which has preventive and therapeutic effects on various diseases such as cerebral ischemia and Alzheimer’s disease [107]. The study showed that the inhibitory effects of cisplatin alone, cyanidin 3-O-glucoside alone and their combination on the proliferation of HeLa cells were 17.43%, 34.98% and 63.38%, respectively. Cyanidin 3-O-glucoside treatment downregulated the PI3K/AKT/mTOR signaling pathway; activated Bax, p53 and tissue inhibitor of metalloproteinase-1 (TIMP-1); decreased cyclin D1 and Bcl-2 expression; and blocked the cell cycle at the G1 phase, thus promoting apoptosis in HeLa cells [108]. Importantly, cyanidin 3-O-glucoside enhanced the sensitivity of HeLa cells to cisplatin. Their combination significantly inhibited the activities of SOD, catalase (CAT) and glutathione peroxidase (GSH-PX) by reducing the expression of NAD(P)H quinone dehydrogenase 1 (NQO1) and heme oxygenase-1 (HO-1), as well as modulating the nuclear factor E2-related factor 2 (Nrf2) signaling pathway to induce oxidative stress and apoptosis in HeLa cells [109]. It is suggested that cyanidin 3-O-glucoside may increase the antitumor activity of cisplatin to reduce the adverse effects associated with chemotherapy in cervical cancer and is a potential strategy for the treatment of cervical cancer.

The chemical structures of flavonoids are summarized in Figure 1. Table 1 summarizes their effects and mechanisms on cervical cancer.

## 4. Terpenoids

### 4.1. Monoterpenoids

Paeoniflorin, a natural monoterpene glycoside compound, is the most important active component of the Chinese medicinal herb *Paeonia lactiflora* and has various protective effects on the cardiovascular system as mediating anti-inflammatory, antioxidant, apoptotic and autophagic regulation [110,111]. Recently, it has been found that paeoniflorin dose- and time-dependently inhibited the proliferation of human endometrial cancer RL95-2 cells by activating p38 MAPK and NF-κB signaling pathways [112]. It also inhibited the proliferation of human gastric cancer MGC-803 cells by inducing apoptosis through upregulation of microRNA-124 (miR-124) and inhibition of PI3K/AKT and transcription activator 3 (STAT3) signaling pathway [113]. In addition, the growth inhibition rate of paeoniflorin on human cervical cancer HeLa cells showed a concentration- and time- dependent relationship, with IC_50_ values of 5054, 2965 and 2459 μg/mL for 24, 48 and 72 h of treatment. It increased Bax/Bcl-2 ratio and mitochondrial outer membrane permeability, caused the release of Cyt c into the cytoplasm, and activated Apaf-1 and caspase 3/9, which induced apoptosis in HeLa cells [114], and it is considered as a potential agent for the treatment of cervical cancer.

Carvacrol is a monoterpene phenol produced by many aromatic plants, including thyme, oregano and other plants. It has various biological and pharmacological properties, such as antibacterial, anticancer, anti-inflammatory, hepatoprotective, antispasmodic and vasodilator properties [115,116]. Carvacrol dose-dependently enhanced cytotoxicity in HeLa cells with IC_50_ value of 556 ± 39 μmol/L after 24 h treatment. In addition, the cytotoxicity of carvacrol was further increased by MEK inhibitor PD325901, which inhibited ERK and increased the levels of LC3β-I/II. Carvacrol induced apoptosis through promoting caspase 9 expression and PARP cleavage [117]. Moreover, carvacrol induced cell-cycle arrest by increasing p53 expression and decreasing cyclin D1 expression in HeLa cells [118]. It may be a promising adjuvant therapeutic agent.

### 4.2. Sesquiterpenoids

In recent years, artemisinin and its derivatives (e.g., dihydroartemisinin and artesunate) have been recognized as effective antimalarial agents [119]. They also have anticancer and anti-angiogenic properties with cytotoxic effects against several cancer types in vitro and in vivo [120,121,122]. Dihydroartemisinin inhibited the proliferation of HeLa and Caski cells in a dose- and time-dependent manner with IC_50_ values of 22.08 and 18.20 μmol/L, respectively. It induced apoptosis through upregulation of Raf kinase inhibitor protein (RKIP) and downregulation of Bcl-2. In a BALB/c mouse model inoculated with HeLa cells or Caski cells, intraperitoneal injection of dihydroartemisinin for 3 weeks significantly inhibited tumor growth with inhibition rates among 70–80% [123]. Moreover, dihydroartemisinin induced apoptosis in HeLa cells and human endometrial cancer cells by decreasing the expression of TfR mRNA and increasing the expression of caspase 3 mRNA. It also led to an increase in LC3-I/II ratio and a decrease in p62 protein levels, which induced autophagic pathway-mediated cell death. The inhibition of the autophagic pathway by using 3-MA further enhanced the cytotoxicity of dihydroartemisinin to HeLa cells [124]. In addition, dihydroartemisinin promoted the production of ROS by upregulating γH2AX protein and foci formation, led to DNA double-strand breaks, while upregulated the phosphorylation of Bcl-2 (Ser70) and mTOR (Ser2448), increased expression of the pro-autophagic protein Beclin-1 and induced autophagic death of Hela cells [125]. Zhang et al. found that dihydroartemisinin significantly inhibited the cell viability of HeLa cells by upregulating the expression of caveolin 1 (Cav1) and mitochondrial carrier homolog 2 (MTCH2) to activate p53 and decrease NQO1 expression, which contributed to the activation of the cell-death pathway and promoted apoptosis in HeLa cells [126].

Artesunate effectively enhanced TRAIL-mediated cytotoxicity by reducing the expression of pro-survival proteins (survivin, XIAP and Bcl-xl) via inhibiting the activation of NF-κB and AKT, and increased apoptosis induced by TRAIL [127]. Besides, artesunate dose-dependently inhibited HeLa and SiHa cell proliferation with IC_50_ values of 5.47 and 6.34 μmol/L, respectively. Interestingly, artesunate in vitro and in vivo increased the radiosensitivity of HeLa cells by promoting apoptosis and G2/M phase transition induced by X-ray irradiation, which was related with the increased expression of cytosolic cyclin B1. It was suggested that it could be applied as an effective radiosensitizer in cancer therapy [128]. Meanwhile, artesunate inhibited cell proliferation by suppressing COX-2 expression in HeLa and CaSki cells, and it decreased PGE2 production and the percentage of CD4^+^CD25^+^Foxp3^+^ T cells [129]. It was reported that subcutaneous injection of 100 mg/kg of artesunate for 15 d effectively reduced the growth and angiogenesis of xenograft tumors by suppressing the secretion of VEGF and the expression of vascular endothelial growth factor receptor KDR/flk-1 on tumors [130]. The results indicate that dihydroartemisinin and artesunate are promising candidates for cervical cancer treatment.

### 4.3. Diterpenoids and Terperpenoids

Tanshinone IIA is a diterpenoid naphthoquinone found in traditional Chinese medicine *Salvia miltiorrhiza*. It has anti-inflammatory and antioxidant activities and has been commonly used in prevention and treatment of cardiovascular disease [131,132]. Radha et al. found that tanshinone IIA caused a significant increase in the expression of p53, p21cip1/waf1, pRb and p130, and activated p53-dependent anticancer activity by inhibiting the expression of HPV E6 and E7 proteins, leading to growth inhibition of cervical cancer cells. In preclinical studies, intraperitoneal injection of 30 mg/kg of tanshinone IIA for 8 weeks significantly reduced the expression of the proliferation marker PCNA in tumor tissues and the volume of cervical cancer transplanted tumors in nude mice by more than 66% [133]. In addition, tanshinone IIA also reduced glycolysis by inhibiting intracellular AKT/mTOR and HIF-1α activities. In vivo, intraperitoneal injection of 40 mg/kg of tanshinone IIA, administered every 2 d for 20 d, effectively inhibited tumor growth and metastasis in U14 cervical cancer mice with an inhibition rate of 72.7% [134]. Another study found that tanshinone IIA inhibited the proliferation of HeLa, SiHa, CaSki and C33A cells in a dose-dependent manner with IC_50_ values of 6.97, 14.47, 5.51 and 9.89 μmol/L respectively, and showed a higher anti-proliferative effect than paclitaxel (18.45 μmol/L). Furthermore, tanshinone IIA treatment activated caspase 9 and 3 to cleave PARP, increased the ratio of Bax/Bcl-2 and JNK, released Cyt c that interacted with Apaf-1, and phosphorylated p38 to trigger ER stress-mediated apoptosis through IRE1 and PERK-related pathways. Notably, tanshinone IIA could improve paclitaxel chemosensitivity, suggesting that it may be a potential strategy to overcome paclitaxel resistance [135]. Importantly, tanshinone IIA inhibited the migration and invasion of cervical cancer stem cells (CSCs) by inhibiting the transfer of HuR from the nucleus to the cytoplasm to reduce the stability and transcriptional activity YAP gene. It was also found that tanshinone IIA not only directly killed the activity of cervical CSCs, but also restored the sensitivity of cervical CSCs to adriamycin [136]. It can be used as an effective therapeutic agent for treating patients with cervical cancer and chemotherapy resistance.

Oridonin is an active diterpene isolated from *Rabdosia rubescens* with various pharmacological and physiological effects, including antibacterial, anti-inflammatory and antitumor [137]. Oridonin as an AKT inhibitor suppressed the growth of cancer cells by attenuating AKT signaling [138]. Hu et al. reported that oridonin inhibited the cell viability of HeLa cells in a dose- and time-dependent manner with an IC_50_ value of 4.13 μmol/L (48 h) [139]. Several studies have shown that oridonin induced apoptosis involving multiple molecular pathways. It significantly inhibited the activity of constitutively activated targets of PI3K (Akt, forkhead box O protein (FOXO) and GSK3) in HeLa cell lines. Treatment of HeLa cells with oridonin activated the cell-death pathway by downregulating Akt kinase signaling, causing loss of MMP to trigger Cyt c release from the mitochondria, leading to downstream activation of caspase 9, caspase 3 and downregulating the expression of the survival proteins (cIAP1, XIAP and survivin), which led to cervical cancer cell death [140,141]. Additionally, ROS is the initiation signal for mitochondrial and caspase-dependent apoptosis as well as autophagy. Oridonin induced ROS production in HeLa cells in a dose-dependent manner, significantly increased Bax and caspase 8 and decreased pro-caspase 3, pro-caspase 9 and Bcl-2, which induced apoptosis and autophagy in HeLa cells [142]. Oridonin also induced apoptosis in HeLa cells through downregulating the level of p-AKT protein and inhibiting the expression and activity of cellular telomerase FKHRL and GSK3β [143]. It may be a key drug in the treatment of cervical cancer.

Ginsenoside Rh2 is a biologically active compound derived from ginseng and has various health effects, including stimulation of immune function, enhancement of cardiovascular health and anti-stress capacity [144]. Ginsenoside Rh2 activated mitochondria-dependent apoptosis pathway and inhibited mitochondrial oxidative phosphorylation and glycolysis in HeLa cells. In addition, ginsenoside Rh2 inhibited energy metabolism and induced apoptosis in HeLa cells by upregulating voltage-dependent anion channel 1 (VDAC1), which caused MMP depolarization and ROS production [145]. It is suggested that VDAC1 is a new target of ginsenoside Rh2. Ginsenoside Rh2 inhibited Hela cell proliferation in a dose- and time-dependent manner by targeting the Akt pathway, and prevented HeLa cell migration and invasion by inhibiting Akt/GSK3β, Snail expression and EMT occurrence [146]. Besides, the combination of ginsenoside Rh2 and betulinic acid induced Bax translocation to mitochondria and released Cyt c, activated caspase 8 and Bid cleavage, and sensitized tumor cells through Bax-dependent mechanism, which decreased cell viability and induced apoptosis [147]. Therefore, ginsenoside Rh2 has the potential to be a novel anticancer drug for cervical cancer.

Betulinic acid, a natural pentacyclic triterpene found in white birch bark, is one of the most promising cancer therapeutic compound with protective effects against tumor progression, inflammation, metabolic diseases and cardiovascular diseases [148,149]. Betulinic acid induced apoptosis in HeLa cells through modulation of PI3K/AKT and mitochondrial pathways. It dose-dependently inhibited the viability of HeLa cells with an IC_50_ value of 30.42 ± 2.39 μmol/L at 48 h of treatment, the phosphorylation of AKT at Thr308 and Ser473, activated Bad, caspase 9 and cell-cycle regulatory factors p27Kip and p21Waf1/Cip1, and induced cell-cycle arrest at G0/G1 phase [150]. Betulinic acid also mediated the accumulation of HIF-1α and inhibited expression of HIF target genes VEGF, GLUT1 and PDK1 in HeLa cells by directly activating proteasome β1, β2 and β5 [151]. Further, betulinic acid decreased Bcl-2 and cyclin D1 and increased the expression of Bax genes, which induced apoptosis and inhibited cell proliferation and migration [152]. It has been shown that betulinic acid is not toxic in vivo at a maximum dose of 500 mg/kg [153]. Betulinic acid is expected to be an adjuvant candidate for human cancer therapy shortly.

The chemical structures of terpenoids are summarized in Figure 2. Table 2 summarizes their effects and mechanisms on cervical cancer.

## 5. Alkaloids

### 5.1. Piperine

Piperine is a nitrogenous stimulant found in the fruits of black pepper and long pepper. It has various pharmacological properties in vitro and in vivo, such as anticancer, antibacterial, antiulcer hepatoprotective and immunomodulatory properties [154,155]. Asif et al. showed that piperine treatment at 50, 100 and 200 μmol/L dose-dependently reduced the cell viability of HeLa cells to 69.90%, 49.27% and 33.54%, respectively. Piperine inhibited HeLa cell proliferation by increasing ROS production and nuclear cohesion, delaying wound healing and blocking the G2/M phase cell cycle. In addition, piperine induced apoptosis in HeLa cells by disrupting MMP and activating caspase-3 [156]. Recently, it has been shown that piperine has the potential to reverse drug resistance in human cervical cancer cells. Piperine (50 μmol/L) downregulated Mcl-1, phosphorylated AKT and cooperated with paclitaxel to increase paclitaxel-induced apoptosis of drug-resistant cervical cancer cells [157]. Apart from this, piperine inhibited Bcl-2 signaling pathway and increased Bid, caspase and PARP activities by blocking STAT3/NF-κB, thereby enhancing therapeutic and antiproliferative effects of mitomycin C (MMC) on drug-resistant HeLa/MMC cells [158]. Consequently, the combination of these two drugs can effectively reduce tumor growth in vivo and is a potential anticancer agent for treating human cervical cancer.

### 5.2. Matrine

*Sophora flavescens* is a kind of Chinese medicine and contains active ingredient matrine. Because of its wide range of pharmacological effects, it has been considered as anticancer drug in China. It has been used clinically to treat viral hepatitis, neuropathic pain, heart disease, skin diseases and other disorders for a long time [159,160]. *S. flavescens* inhibits cervical-cancer-cell proliferation and metastasis and induces apoptosis by inhibiting the AKT/mTOR signaling pathway; reducing the expression of Bcl-2, cyclin A and MMP2; blocking the G2/M phase cell cycle; and stimulating Bax and E-calmodulin. Matrine reduces vasodilator-stimulated phosphoprotein (VASP) phosphorylation by inhibiting protein kinase A (PKA) activity, thereby inhibiting HeLa cell adhesion and migration [161]. Besides, in mice injected with HeLa cells, intraperitoneal injection of matrine with 50 mg/kg/d for 21 d effectively inhibited the growth of cervical cancer xenografts with an inhibition rate of 58.33%. It also reduces the expression activity of extracellular matrix factors MMP2 and MMP9 by downregulating the p38 signal pathway, thus inhibiting the proliferation, metastasis and invasion of cervical cancer cells [162]. Matrine has the potential for development as therapeutic or adjuvant agent for human cervical cancer.

### 5.3. Berberine

Berberine, an isoquinoline alkaloid extracted from the herb, has various biological activities, such as antibacterial, anti-inflammatory, antidiabetic, anti-angiogenic and anticancer effects. It is widely used as an antibacterial agent in clinical practice [163,164]. Berberine inhibited the proliferation of HeLa and SiHa cells in a dose-dependent manner. It reduced cell viability and hTERT expression through a mitochondria-mediated pathway and activated caspase3, thereby inducing apoptosis. Moreover, berberine can selectively inhibit the transcription factor Activator Protein-1 (AP-1) activity and the transcription of HPV oncoproteins E6 and E7 and increase the expression of p53 and Rb to play an anti-cervical-cancer effect [165]. A study showed that berberine inhibited the invasion of SiHa cells by reducing the transcriptional activity of MMP2 and u-PA, mediated the downregulation of TGF-β1 expression and inhibited the angiogenic potential of SiHa cells. Notably, berberine (20 mg/kg/d for 21 d by oral gavage) was able to downregulate the NF-κB signaling pathway, increase E-calmodulin expression and selectively inhibit snail-1 gene expression, thereby reversing EMT and reducing cancer cell growth and lung metastasis [166]. It was also found that the IC_50_ value of berberine was 300 μmol/L after the 48-h treatment of HeLa cells. Berberine activated the death receptor pathway in HeLa cells by upregulating Fas, FasL, TNF-α and TRAF-1. Simultaneously, berberine induced apoptosis in HeLa cells by activating the mitochondrial pathway, releasing Cyt c and significantly increasing the ratio of Bax/Bcl-2, which blocked the cell cycle at the S phase. Moreover, berberine activated the MAPK pathway and increased p53 expression, suggesting that p53 may be a drug target in the treatment of cervical cancer [167]. In addition, N-Mannich bases of berberine (methoxy functional group, acid functionality and cyano group), berberine analogs, had significant cytotoxicity in cervical cancer HeLa and CaSki cells [168]. It suggested the potential antitumor effect of berberine and its analogs on cervical cancer cells.

The chemical structures of alkaloids are summarized in Figure 3. Table 3 summarizes their effects and mechanisms on cervical cancer.

## 6. Phenols

### 6.1. Curcumin

Curcumin is a phenol compound extracted from the rhizome of the perennial herb turmeric in Zingiberaceae, and has been commonly used as a spice and food coloring. Its safety and efficacy as a traditional herbal medicine have been shown in previous studies. It also exerts anticancer effects through various mechanisms, including inhibition of cell proliferation, invasion and metastasis, and epigenetic alterations [169,170]. Curcumin directly inhibited telomerase activity in HeLa, SiHa and CaSki cells in a dose-dependent manner, induced Cyt c release, activated AIF and inhibited cyclin D1 and Ras proteins, thereby downregulating downstream targets of the Ras/Raf pathway and ultimately triggering apoptosis via the mitochondrial pathway. Curcumin also mediates the anti-proliferative and anti-inflammatory activities of cervical cancer cells by inhibiting COX-2 and inducible nitric oxide synthase (iNOS) expression and activating caspase 3 and caspase 9 [171]. It has been reported that oral gavage of curcumin (500, 1000 and 1500 mg/kg/d) for 30 d in a mouse model injected with CaSki cells significantly inhibited tumor-cell growth (21.03%) and the expression of VEGF and EGFR, which inhibited angiogenesis [172]. In addition, curcumin inhibited the proliferation of HeLa, CaSki, C33A and ME180 cells in a time- and concentration-dependent manner; and it activated ER-resident unfolded protein response (UPR) sensors (PERK, IRE-1a and ATF6) and their downstream signaling proteins in cervical cancer cells, especially CHOP. Activation of CHOP elevated the Bax/Bcl-2 ratio and increased ROS production, which induced apoptosis in cervical cancer cells [173]. On the other hand, curcumin induced G2/M-phase cell-cycle arrest and triggered cell autophagy and apoptosis through downregulation of cyclin B1 and Cdc25. At the same time, it mediated cellular senescence by elevating the p53/p21 pathway [174]. Furthermore, curcumin treatment effectively reduced P-gp expression and increased the sensitivity of drug-resistant KB-V1 cells to vinblastine [175]. To enhance the bioavailability of curcumin, Warayuth et al. developed a delivery vehicle for oral curcumin. The cytotoxicity test showed that the IC_50_ values of drug-loaded (curcumin) micelles on HeLa, SiHa and C33a cell lines were 4.7, 3.6 and 12.2 times lower than those of free curcumin (21.17, 16.28 and 54.29 μmol/L). Compared with free curcumin, the number of micelles carrying curcumin is significantly increased by 6-fold. It is worth noting that the percentage of early apoptosis of HeLa, SiHa and C33a cells is increased by curcumin micelles to 30–55% [176]. These data provide a possible theoretical basis for clinical application of cervical cancer.

### 6.2. Ellagic Acid

Ellagic acid is a natural phenol compound primarily found in fruits and nuts, including raspberries, pomegranates and walnuts, with strong anti-proliferative, anti-metastatic and anti-angiogenic activities [177,178]. Ellagic acid inhibited SiHa cell proliferation in a dose-dependent manner with an IC_50_ value of 48.7 ± 2.5 μmol/L. Ellagic acid promoted SiHa cell apoptosis by upregulating p53, Bcl-2 and caspase 3/9; downregulating Bax; and inducing the G2-phase cell-cycle arrest [179]. Li et al. reported that ellagic acid induced G1-phase cell-cycle arrest by inhibiting the activation of janus kinase 2 (JAK2)/STAT3 pathway and downregulating the expression of Cyclin D1, Bcl-xl and Mcl-1, which in turn promoted apoptosis [180]. Moreover, ellagic acid inhibits the AKT/mTOR signaling pathway by increasing insulin-like growth factor-binding protein 7 (IGFBP7), thereby suppressing HeLa cell proliferation and invasion [181]. A recent study showed that treatment of HeLa cell with ellagic acid, curcumin and their combination for 72 h resulted in IC_50_ values of 19.47, 16.52 and 10.9 μmol/L, which revealed the synergistic anticancer effect of ellagic acid and curcumin. Their combination synergistically restored p53 protein expression, induced ROS formation and increased DNA damage, accompanied by increased expression of the downstream effectors p21 and Bax, which induced apoptosis in HeLa cells [182]. Ellagic acid has the potential to be an ideal drug for the treatment of cervical cancer.

### 6.3. Resveratrol

Resveratrol is a dietary polyphenol derived from grapes, berries, peanuts and other plant sources. It has several potential biological effects, including cancer prevention and treatment [183]. Resveratrol inhibited the proliferation of HeLa and CaSki cells in a dose- and time-dependent manner with IC_50_ values of 40.06 and 59.07 μmol/L at 72 h. Resveratrol inhibited HeLa and CaSki cell growth and development by inhibiting the expression of HPV oncoproteins E6 and E7; promoting p53, Bax and p16; and inducing cell-cycle arrest at the G1/S phase [184]. A study proved that resveratrol inhibited cervical cancer cell proliferation by upregulating suppressors of cytokine signaling 3 (SOCS3) and activated STAT3 (PIAS3) expression, blocking the DNA-binding activity of STAT3 and causing STAT3 inactivation [185]. Furthermore, resveratrol suppressed the degradation of EMT and extracellular matrix (ECM) by inhibiting STAT3 (Tyr705) phosphorylation [186]. He et al. found that resveratrol induced apoptosis in HeLa cells by downregulating hTERT mRNA and protein levels, which in turn inhibited telomerase activity [187]. Importantly, resveratrol also inhibited the expression of HIF-1α and VEGF in cervical cancer cells by blocking ERK1/2 and PI3K/Akt signaling pathways, thereby effectively reversing the angiogenic activity induced by HPV-16 E6 and E7 oncoproteins [188]. Another experiment proved that intraperitoneal injection of 10 mg/kg of resveratrol for 28 d significantly inhibited the growth of xenograft tumors in nude mice. Resveratrol significantly decreased the mRNA and protein expression of phospholipid scramblase 1 (PLSCR1), thereby inhibiting the proliferation and development of HeLa cells. In addition, resveratrol also inhibited migration and invasion of HeLa cells by suppressing NF-κB and AP-1-mediated MMP9 expression [189]. Resveratrol offers potential therapeutic value for the treatment of cervical cancer.

The chemical structures of phenols are summarized in Figure 4. Table 4 summarizes their mechanisms in cervical cancer. Figure 5 outlines the mechanism of natural products against cervical cancer

## 7. Conclusions

The high mortality and morbidity rates of cervical cancer remain a primary challenge for scientific research. Although conventional treatments, such as radiotherapy and chemotherapy, are effective, the five-year survival rate of cervical cancer patients is meager and often ends in failure. Moreover, chemotherapy is prone to drug resistance and toxic side effects, causing great suffering to patients. In recent years, plant-derived natural products have been considered as the most promising candidates for oncology therapies. In 2010, Bar-Sela et al. [190] reported a review article on curcumin as an anticancer agent, reviewing and analyzing data from clinical trials focusing on colon and pancreatic cancer and summarizing the main anticancer mechanisms of curcumin, but the article was published relatively early, ten years ago. Therefore, there is a need for further exploration and systematic review and sorting of the latest effective antitumor studies. In 2019, Hsiao et al. reviewed the effects and potential mechanisms of Chinese herbal medicines on cervical cancer and classified them according to crude extracts and compounds, outlining their effects on induction of apoptosis and inhibition of migration in vitro and in vivo, but the number of compounds was small [191]. In 2021, Park et al. reviewed the therapeutic effects of a variety of natural products on cervical cancer and their mechanisms, including apoptosis, anti-angiogenesis, anti-metastasis, drug resistance and microRNA modulation. However the inhibition of telomerase activity and enhancement of immune function were not included [192]. Our study reviewed 30 natural products that have antitumor effects on cervical cancer, which showed possible benefits in treating patients with cervical cancer through mechanisms, such as induction of apoptosis, inhibition of cell proliferation and angiogenesis (Figure 6). Therefore, the search for more readily available natural antitumor active ingredients and precursor drugs could provide effectively alternative or adjuvant treatment strategies for cancer patients.

Plant-derived natural products, a rich treasure trove of drug resources, have driven the development of novel inhibitors with their unique mode of action and molecular structural diversity. They play an important role in human cancer therapy with high safety and low side effects. However, some unresolved issues limit the translation of plant-derived natural products to clinical applications. First, some have poor water solubility, which seriously affects bio-availability. Second, potential toxicity and adverse effects of plant-derived natural products are lack of sufficient evaluation in vivo and clinical trial data. Again, further research on the mechanism of natural antitumor drugs is still the direction of future efforts. We believe that, with additional research, plant-derived natural products will be indeed applied in clinical medicine.

## Figures and Tables

**Figure 1 biomolecules-11-01539-f001:**
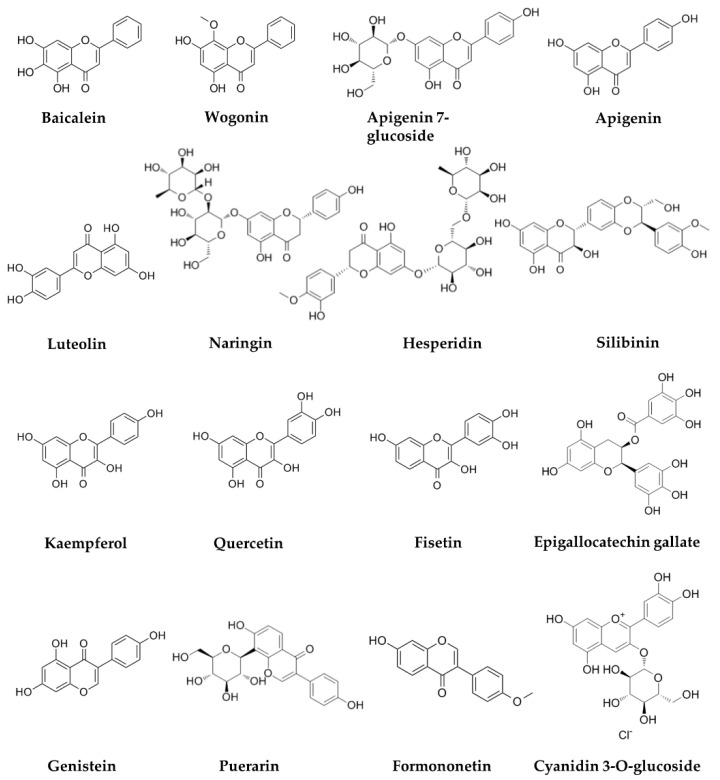
Chemical structure of several flavonoids against cervical cancer.

**Figure 2 biomolecules-11-01539-f002:**
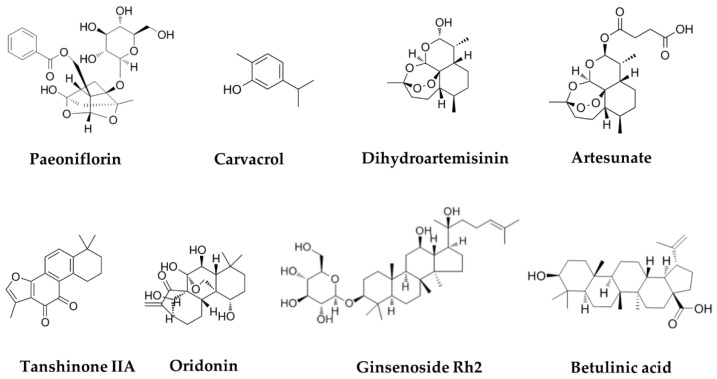
Chemical structure of several terpenoids in anti-cervical-cancer.

**Figure 3 biomolecules-11-01539-f003:**
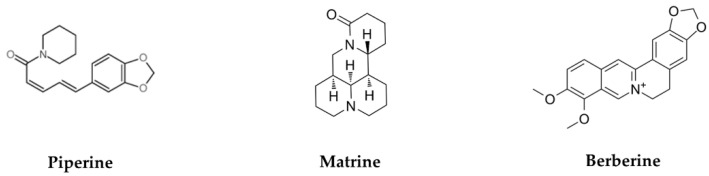
Chemical structure of several alkaloids in anti-cervical-cancer.

**Figure 4 biomolecules-11-01539-f004:**
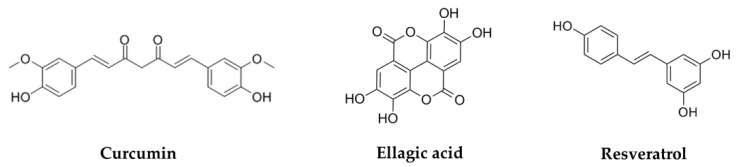
Chemical structure of several phenols against cervical cancer.

**Figure 5 biomolecules-11-01539-f005:**
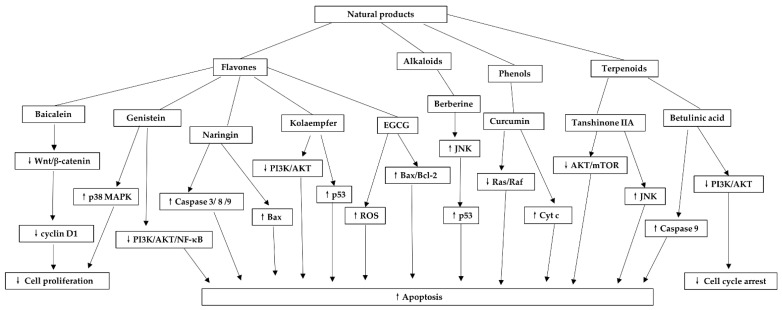
Mechanism of natural products against cervical cancer. Baicalein inhibits Cyclin D1 expression through downregulation of the Wnt/β-catenin pathway, which in turn inhibits cell proliferation. Genistein inhibits cell proliferation by activating the p38 MAPK pathway. It also induces apoptosis by inhibiting the PI3K/AKT-NF-κB pathway, which in turn induces apoptosis. Naringin promotes apoptosis by up-regulating the expression of Bax and caspase 3/8/9, which in turn promotes apoptosis. Kaempferol is pro-apoptotic by inhibiting PI3K/AKT and activating the p53 pathway. EGCG induces apoptosis by increasing ROS production and Bax/Bcl-2 expression. Berberine induces apoptosis by activating JNK and p53, while Curcumin promotes apoptosis by inducing Cyt c release, thereby downregulating the Ras/Raf pathway. Tanshinone IIA induces apoptosis through inhibition of AKT/mTOR and activation of the JNK pathway. Betulinic acid induces cell cycle arrest and apoptosis by activating caspase 9 and inhibiting the PI3K/AKT pathway. ↑: Upregulation, ↓: Downregulation.

**Figure 6 biomolecules-11-01539-f006:**
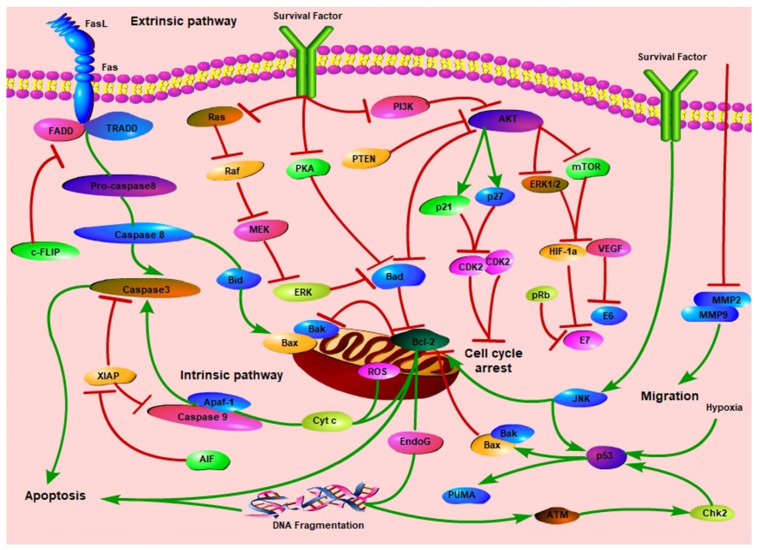
Overview of signaling pathways against cervical cancer.

**Table 1 biomolecules-11-01539-t001:** Effect and mechanism of flavonoids on cervical cancer.

Chemical Family	Molecule Name	Concentration	Cell Line	Mechanism
Flavones	Baicalein	100 μmol/L	HeLa	Increased: activation of caspase 3; PARP cleavage Decreased: cIAP-1, cIAP-2, FLIP, Bcl-2, MMP2, MMP9, caspase 8, Fas, FasL, VEGF, COX-2, cyclin D1, IL-8, MCP1; Inhibit NF-κB, ERK1/2; G1 phase cell block
		10 mg/kg for 28 d	Xenografts of cervical cancer HeLa cells in female BALB/c mice	Decreased: tumor growth
	Wogonin	0–100 μmol/L	SiHa and CaSki	Increased: Bax; activation of caspase 3 and 9; Cyt c release; PARP cleavage Decreased: MMP; Bcl-2
		40–160 μmol/L	SiHa and CaSki	Increased: Bax, p53, p21, p27, pRb Decreased: E6, E7
	Apigenin 7-glucoside	IC_50_ = 47.26 μmol/L	HeLa	Increased: ROS, Fas, FasL, TNF-α, TNF-r1, FADD, RADD; activation of caspase 3 and 9 Decreased: Bcl-2, Bcl-xl, Cyclin (A, D, E), CDK2/6, MMP; Inhibit PTEN/PI3K/AKT;
	Apigenin	IC_50_ = 10, 68, 76 and 40 μmol/L	HeLa, SiHa, CaSki and C33A	Increased: ROS, H_2_O_2_ Decreased: MMP
		100 mg/kg for 30 d	Xenografts of cervical cancer HeLa cells in female BALB/c mice	Decreased: ERβ/ERα, tumor growth
	Luteolin	IC_50_ = 21.8 μmol/L	HeLa	Increased: p16 ^INK4A^, JNK Decreased: UHRF1, DNMT1, A20, c-IAP1; G2/M phase cell block
Flavanones	Naringin	IC_50_ = 750 μmol/L	SiHa	Increased: p53, Bax, Fas, FADD; activation of caspase 3, 8 and 9
	Hesperidin	0–100 μmol/L	HeLa	Increased: AIF, Cyt c release; activation of caspase 3 Decreased: MMP, cyclin D1, cyclin E1, CDK2; G0/G1 phase cell block
	*Silibinin*	IC_50_ = 250,195 μmol/L	HeLa and SiHa	Increased: activation of Drp1 Decreased: ATP, MMP, ROS, CDK1, Cdc25C, cyclinB1; G2/M phase cell block
Flavonols	Kaempferol	IC_50_ = 10.48 μmol/L	HeLa	Increased: Bax, p53 Decreased: Bcl-2, hTERT; Inhibit PI3K/AKT
	Quercetin	IC_50_ = 110.38 ± 0.66 μmol/L	HeLa	Increased: Bax, p53, ROS; activation of caspase 3; Cyt c release Decreased: Bcl-2, AKT, MMP; G2/M cell block
		0–200 μmol/L	HeLa and SiHa	Increased: Bax, p53, p21 Decreased: E6/E6AP, G2 phase cell block
	Fisetin	IC_50_ = 36.0 ± 0.5 μmol/L	HeLa	Increased: ERK1/2, activation of caspase 3 and 8
		2–4 mg/kg for 35 d	Xenografts of cervical cancer HeLa cells in male BALB/c mice	Decreased: tumor growth, with inhibition rates of 82.65% and 92.62%
Flavanols	Epigallocatechin gallate	IC_50_ = 20 μmol/L	HeLa	Increased: Bax, p53, ROS; Cyt c release Decreased: Bcl-2, COX-2; inhibition AKT and NF-κB
Isoflavones	Genistein	IC_50_ = 20 and 60 μmol/L	HeLa and CaSki	Increased: p38 MAPK, p38-JNK Decreased: ERK1/2, AKT
		20 mg/kg	C57BL/6 cervical cancer cell mice model	Decreased: tumor growth
	Puerarin	12.5–50 μmol/L	HeLa	Increased: Bax Decreased: Wnt/β-catenin, p21, p53, Bcl-2
		0–2000 μmol/L	HeLa	Increased: Bax Decreased: Inhibit PI3K/AKT/mTOR
		500 mg/kg for 15 d	cervical cancer cell line U14 mice models	Increased: IL-2, SOD Decreased: tumor growth
	Formononetin	0–100 μmol/L	HeLa	Increased: Bax, ROS; activation of caspase 3 and 9 Decreased: Bcl-2, MRP1 and MRP2, MMP
		20 and 40 mg/kg for 35 d	Xenografts of cervical cancer HeLa cells in BALB/c nude mice	Decreased: tumor growth
Anthocyanins	Cyanidin 3-O-glucoside	400 μmol/L	HeLa	Increased: Bax, p53, TIMP-1 Decreased: Bcl-2, cyclin D1; G2/M phase cell block; PI3K/AKT/mTOR

**Table 2 biomolecules-11-01539-t002:** Effect of terpenoids on cervical cancer.

Chemical Family	Molecule Name	Concentration	Cell Line	Mechanism
Monoterpenoids	Paeoniflorin	IC_50_ = 2459 μg/mL	HeLa	Increased: Bax, Apaf-1; activation of caspase 3; Cyt c release Decreased: Bcl-2
	Carvacrol	IC_50_ = 556 ± 39 μmol/L	HeLa	Increased: LC3β-I/II; activation of caspase 9; PARP cleavage Decreased: ERK
Sesquiterpenoids	Dihydroartemisinin	IC_50_ = 22.08 and 18.20 μmol/L	HeLa and Caski	Increased: RKIP Decreased: Bcl-2
		20 μmol/L for 15 d	Xenografts of cervical cancer HeLa or Caski cells in BALB/c mice	Decreased: tumor growth, with inhibition rates of 70–80%
	Artesunate	60 μg/mL	HeLa	Increased: AKT; activation of caspase 3; PARP cleavage Decreased: survivin, XIAP, Bcl-xl; G2/M phase cell block; Inhibit NF-κB
		100 mg/kg for 15 d	Xenografts of cervical cancer HeLa cells in BALB/c nude mice	Decreased: tumor growth and inhibition of angiogenesis
Diterpenoids	Tanshinone IIA	IC_50_ = 6.97, 14.47, 5.51, and 9.89 μmol/L	HeLa, SiHa, CaSki and C33A	Increased: Bax, PERK, IRE1, p38, JNK; activation of caspase 3 and 9; PARP cleavage; Cyt c and Ca^2+^ release Decreased: Bcl-2
		0–10 μmol/L	HeLa, SiHa, CaSki	Increased: p53, p21, p130, pRb Decreased: E6, E7
		40 mg/kg for 20 d	Cervical cancer cell line U14 mice models	Decreased: metastasis and tumor growth with inhibition rates of 72.7%
	Oridonin	C_50_ = 4.13 μmol/L	HeLa	Increased: Bax; activation of caspase 3 and 9; Cyt c release Decreased: Bcl-2, MMP
Triterpenoids	Ginsenoside Rh2	C_50_ = 35 μmol/L	HeLa	Increased: Bax, ROS, VDAC1; Cyt c release Decreased: MMP
	Betulinic acid	C_50_ = 30.42 ± 2.39 μmol/L	HeLa	Increased: Bad, ROS, p27^Kip^ and p21^Waf1/Cip1^; activation of caspase 9 Decreased: G0/G1 phase cell block; Inhibit PI3K/AKT

**Table 3 biomolecules-11-01539-t003:** Effect of alkaloids on cervical cancer.

Chemical Family	Molecule Name	Concentration	Cell Line	Mechanism
Alkaloids	Piperine	0–200 μmol/L	HeLa	Increased: ROS; activation of caspase 3 Decreased: MMP; G2/M phase cell block
	Matrine	50 mg/kg for 21 d	Xenografts of cervical cancer HeLa cells in BALB/c nude mice	Decreased: p38, MMP2 and 9; tumor growth with inhibition rates of 58.33%
	Berberine	IC_50_ = 300 μmol/L	HeLa	Increased: Bax, Fas, FasL, TNF-α, TRAF-1, p53, MAPK; DNA fragmentation; activation of caspase 3 Decreased: Bcl-2; S phase cell block
		0–250 μg/mL	HeLa and SiHa	Increased: p53, pRb Decreased: E6, E7, AP-1, c-Jun, c-Fos

**Table 4 biomolecules-11-01539-t004:** Effect of phenols on cervical cancer.

Chemical Family	Molecule Name	Concentration	Cell Line	Mechanism
Phenols	Curcumin	50 and 100 μmol/L	HeLa, SiHa and CaSki	Increased: AIF; activation of caspase 3 and 9; Cyt c release Decreased: COX-2, iNOS, cyclin D1; Inhibit Ras/Raf
		500, 1000 and 1500 mg/kg	Xenografts of cervical cancer CaSki cells in BALB/c mice	Decreased: tumor growth, VEGF and EGFR
	Ellagic acid	C_50_ = 48.7 ± 2.5 μmol/L	SiHa	Increased: Bcl-2, p53; activation of caspase 3 and 9 Decreased: Bax; G2 phase cell block
	Resveratrol	C_50_ = 40.06, 59.07 μmol/L	HeLa	Increased: Bax, p53, p16 Decreased: G1/S phase cell block
		5–250 μmol/L	HeLa and CaSki	Decreased: Inhibit PI3K/AKT, ERK1/2, VEGF, HIF-1α accumulation
		10 kg/mg for 28 d	Xenografts of cervical cancer HeLa cells in nude mice	Decreased: tumor growth

## Data Availability

Not applicable.

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
