# Peer review of "Potential Mechanisms of Plant-Derived Natural Products in the Treatment of Cervical Cancer"

_biomolecules, 2021, doi:10.3390/biom11101539_

Round 1

Reviewer 1 Report

Authors reviewed the potential mechanisms of natural compounds. The natural compounds were divided in to their classification. The quality of the study is low and there are parts should be revised or added.

  1. The figures that represent the mechanisms of natural compounds should be added in result and discussion.
  2. In discussion section, the overall analysis, strength and limitations of this review should be added.  
  3. There are similar review articles published like PMID: 33466408, PMID: 31271066 , PMID: 20214562. The comparison is needed in discussion.
  4. Why did authors include lignans or stilbenes which are member of polyphenol.
  5. What are the keywords, search engines used, exclusion criteria? please add methods section for those information.
  6. HPV is theprimary cause of cervical cancer, the natural product that regulate the HPV andits oncoproteins should be organized as table and their mechanisms should be schematized as a figure.

Reviewer 2 Report

Meizhu He et al., wrote a review on natural products as anti-cervical cancer agents and explain multiple mechanisms to reversal the multidrug resistance. Additionally, the therapeutic effects and mechanisms of plant-derived natural products on cervical cancer.

This review well written as idea but need some modification in its presentation. For example: 2. Flavonoids and a subsection flavones followed by plants extracts and compounds (as major metabolites). I   suggest to be the following for all review items:

1- introduction

2- Major secondary metabolites

2.1 flavonoids

2.1.1. flavones

2.1.1.1 Compound

2.1.1.2 compound

2.2 monoterpenes

2.3. sesquiterpenes……….

3- plant extracts

3.1 Green tea

3.2. …………

Additionally, the review should include a figure summarize the most potent compounds and its cells (or invivo studies)  and/or mechanisms to be easy for readers.

Reviewer 3 Report

In recent years, plant-derived natural components such as flavonoids, terpenoids and alkaloids have been considered as the most promising products for oncology therapies through multiple mechanisms, including inhibition of tumour cell proliferation, induction of apoptosis, suppression of angiogenesis and telomerase activity, enhancement of immunity, and reversal of multidrug resistance.

This paper reviews the latest studies on the role and mechanisms of plant-derived natural products in the treatment of cervical cancer. The manuscript is well written, treats an actual problem and provides readers with the highlights on the topic. The title is appropriate for the content of the article. The abstract is concise and accurately summarizes the essential information of the paper. The introduction provides a good, generalized background of the topic that gives the reader relevant background information.

Prior to publication, only few minor corrections should be made:

Line 89, 110, 128, 208, 244, 271, 383, 405, 408, 448, 550: Please change to IC50

Line 152: Silybum marianum (italic)

Line 228: Please change μmol / L to μmol/L

Line 312: Please change Anthocyandins to Anthocyanins, because the components presented in this section are mainly anthocyanins

Table 1: Please change Anthocyandins to Anthocyanins

Line 334: Please change Monoterpenes to Monoterpenoids (the compounds presented in this section are mainly monoterpenoids

Line 359: Please change Sesquiterpenes to Sesquiterpenoids (dihydroartemisinin and artesunate presented in this section are sesquiterpenoids

Line 393: Please change Diterpenes to Terperpenoids (the compounds presented in this section are mainly diterpenoids

Line 394: Please change “Tanshinone IIA is a diterpene naphthoquinone” to “Tanshinone IIA is a diterpenoid naphthoquinone” and “salvia” to “Salvia

Figure 1, 2 and 3 are not mentioned in the all text

Table 1, 2 and 3: Please add “of” in the title; for example “Effect of flavonoids” end etc.

Section 6 “Discussion”: Isn't that the conclusion?

Round 2

Reviewer 1 Report

The manuscript is revised. However, the number of the articles contained is few which is 30. There are much more articles could be found using ‘natural product, cervical cancer’ as keywords. What are the exclusion criteria? The search engines used were only PUBMED and Google Scholar. Biolmolecules is high impact factor Journal. More information should be reviewed systemically.

Reviewer 2 Report

The manuscript is accepted